# MSCoRe: A Benchmark for Multi-Stage Collaborative Reasoning in LLM Agents

## Abstract

Large Language Model (LLM) agents have excelled in single-stage tasks. However, their reasoning and coordination capabilities in multi-stage scenarios remain underexplored. Existing benchmarks typically focus on isolated tasks or narrow domains, overlooking models' abilities for multi-stage collaboration and optimization without explicit external guidance. To bridge this gap, we propose **MSCoRe**, a novel benchmark comprising 126696 domain-specific QA instances spanning scenarios in automotive, pharmaceutical, e-commerce, and automotive energy sectors. We also introduce a structured three-phase pipeline: dynamic sampling, iterative question-answer generation, and a multi-level quality assessment to generate high-quality data. For a more refined assessment, we categorize tasks into three difficulty levels based on their stage coverage and complexity. With MSCoRe, we have conducted a comprehensive evaluation of various state-of-the-art LLM agents. The commercial models performed best across all tasks and scenarios, but a notable gap in the ROUGE scores remains between simple and complex tasks. We also tested the models' robustness under three types of noisy data and found that their performance is negatively affected by different noise. MSCoRe provides a new resource for evaluation and improvement of multi-stage collaborative reasoning in LLM agents. Codes and data are available at `https://huggingface.co/datasets/032564yn/MSCoRe`.

## 1 Introduction

Reasoning tasks serve as a crucial assessments for evaluating the advanced capabilities of Large Language Model(LLM) agents. It requires both precise intent understanding and the generation of coherent, logically sound responses. General purpose benchmarks such as WikiQA (Yang et al., 2015), MMLU (Hendrycks et al., 2020), CommonsenseQA (Talmor et al., 2018), and ZeroShot (Sun et al., 2025) span diverse domains and provide a rigorous evaluation of the foundational reasoning and knowledge of the LLM agents. As models became more sophisticated, benchmarks like HotpotQA (Yang et al., 2018) and QASC (Khot et al., 2020) evolved to test complex reasoning capabilities. Currently, domain-specific datasets such as EQUALS (Chen et al., 2023a) for law, DISC-FinLLM (Chen et al., 2023b) for finance, and MedExQA (Kim et al., 2024) for medicine have emerged to assess specialized expertise. However, a limitation persists across these evaluations: they treat tasks as isolated, single-stage problems, overlooking the collaborative reasoning required between different stages in complex domains.

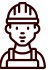

How can reducing vehicle weight improve fuel efficiency ?

**Single-Stage Reasoning**

Lightweight materials and structural optimization.

- - - - - - - - - - - - - - - - - - - - - - - - - -

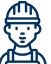

How to improve fuel efficiency through vehicle design and manufacturing ?

**Multi-Stage Reasoning**

- **Simulation Analysis:**
    CFD Fluid Simulation
    FEA Structural Analysis
- **Material Selection:**
    Carbon Fiber Composite Optimization
- **Manufacturing Processes:**
    Advanced Joining Technologies
    Heat Treatment Processes

Figure 1: Comparison of single-stage reasoning and multi-stage reasoning approaches in the automotive design problem-solving.

This limitation is particularly pronounced in real-world industrial applications where workflows are often multi-staged. For example, the automotive value chain involves interconnected stages

from design and manufacturing to supply chain and quality inspection (Pinho Santos & Proenca, 2022). A design choice directly impacts manufacturing feasibility, while a supply chain disruption can halt production. Similarly, in the pharmaceutical industry, quality inspection results provide essential feedback for optimizing production processes (Glória et al., 2024). Constructing agents in specialized domains demands not only domain expertise but also collaborative reasoning abilities that consider downstream consequences and upstream constraints. Existing evaluation frameworks struggle to measure this crucial holistic problem-solving ability.

To bridge this gap, we introduce **M**ulti-**S**tage **Co**llaborative **Re**asoning (MSCoRe) benchmark, a novel benchmark designed to assess the multi-stage collaborative reasoning of LLM agents. Different from existing benchmarks, MSCoRe foucus on collaborative reasoning tasks within multi-stage scenarios. Figure 1 shows an example of comparison between multi-stage collaborative reasoning and single-stage knowledge question-answering task. MSCoRe comprises 126696 high-quality QA instances spanning four domains: Automotive Value Chain, Pharmaceutical Value Chain, E-Commerce Value Chain, and Automotive Energy chain. Inspired by automated generation frameworks (Goyal et al., 2020; Wang et al., 2022b; Madaan et al., 2023), we propose a structured pipeline that combines high-quality seed data sampling, refined prompt engineering, and a multi-level quality control mechanism with feedback-driven optimization. For more detailed analysis, we categorize tasks into three difficulty levels: simple, medium, and complex, based on their complexity and the scope of stages involved.

With MSCoRe, we have conducted a comprehensive evaluation of various state-of-the-art LLM agents, including the commercial models like GPT-4 (Achiam et al., 2023), GPT-3.5-turbo, and Claude-3.5-haiku, as well as open-source models like DeepSeek-R1 (Guo et al., 2025), and Qwen series (Bai et al., 2023). The results suggest that models that excel at simple tasks still face significant challenges when it comes to complex tasks. Furthermore, we introduce three forms of noise into data, including formatting errors, missing information, and semantic incompleteness, to assess robustness and interference resistance of models under adverse input conditions.

In summary, the contributions of this paper are summarized as follows:

- We construct MSCoRe, a novel large-scale benchmark with 126696 QA instances across four domains. The benchmark fill the gap in the evaluation of multi-stage collaborative reasoning of LLM agents.

- We propose a structured pipeline for the automated generation of high-quality question-answering data within specialized domains, and through refined prompts, we generate QA instances of various difficulty levels.

- We comprehensively evaluate the performance of the state-of-the-art LLMs on MSCoRe, highlight the weakness of current LLM agents in multi-stage collaborative reasoning. MSCoRe provides a rich data resource for evaluating and improving the multi-stage collaborative reasoning capabilities of future LLM agents.

## 2 RELATED WORK

### 2.1 BENCHMARKS FOR REASONING ABILITY

The rapid evolution of Large Language Models (LLMs) has been accompanied by the development of benchmarks to evaluate their various capabilities. Reasoning benchmarks explore diverse capabilities of LLMs from multiple perspectives. Firstly, some benchmarks primarily focus on evaluating models' general capabilities, such as GLUE (Wang et al., 2018), SuperGLUE (Sarlin et al., 2020) and MMLU (Hendrycks et al., 2020), which established a foundation for evaluating natural language understanding. However, as LLMs grew in scale and complexity, more extensive evaluation suites became necessary. Benchmarks such as BIG-bench (Srivastava et al., 2023) tests for a diverse set of emergent abilities. HELM (Liang et al., 2022) advocate for a multi-metric, holistic evaluation. Lately, the evaluation focus has shifted towards more interactive and agent-like behaviors. For instance, AgentBench (Liu et al., 2023) evaluates LLMs in complex, multi-turn open-ended environments, and ToolBench (Qin et al., 2023) assesses their ability to use external tools to solve problems. Although these benchmarks are crucial for understanding the overall reasoning and task execution capabilities of LLMs, they typically consist of a series of independent tasks, which fail to capture the complex causal dependencies inherent in multi-stage workflows. MSCoRe is specifically designed to address this gap by focusing on these cross-stage relationships.

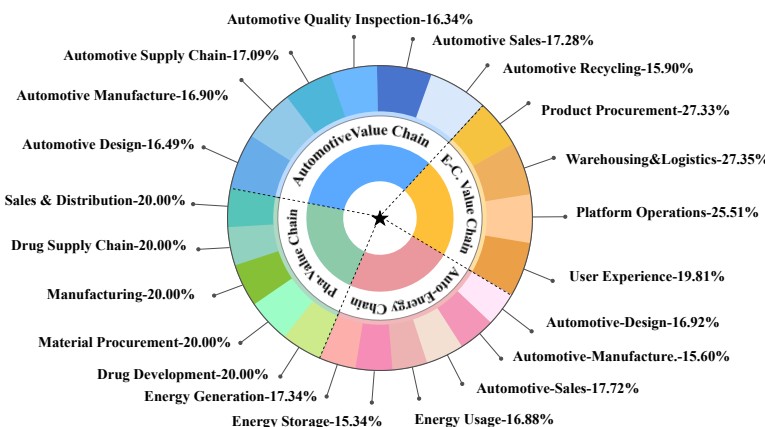

Figure 2: Inner layer: Domain of data distribution, outer layer: work stages contained within the domain; numbers: percentage of data from each stage within the domain's dataset. MSCoRe covers 4 domains: Automotive Value Chain, E-Commerce Value Chain, Automotive Energy Chian, and Pharmaceutical Value Chain.

## 2.2 DOMAIN-SPECIFIC LLM BENCHMARKS

Recognizing that real-world applications require specialized knowledge, there has been a significant trend towards creating domain-specific benchmarks. In fields like medicine, benchmarks such as MedQA (Jin et al., 2021) and MedMCQA (Kim et al., 2024) evaluate clinical knowledge by testing models on medical exam questions. The recent evaluation framework for Med-PaLM 2 further highlighted the need for rigorous assessment against real-world medical challenges (Singhal et al., 2025). Similarly, the finance sector has seen the introduction of benchmarks like FinQA (Chen et al., 2023a) and the more recent FinEval (Zhang et al., 2023), which focus on financial report analysis and economic knowledge. In the legal domain, benchmarks like LexGLUE (Chalkidis et al., 2021) test models on legal text processing and reasoning tasks. These domain-specific benchmarks are indispensable for validating the factual accuracy and specialized knowledge of LLMs. However, their primary focus is often on assessing a model's ability to act as a knowledge repository or a domain expert. For example, they might test if a model can answer a specific legal question or interpret a financial table. They generally do not evaluate a model's capacity to reason through a multi-stage process, such as managing the automotive supply chain from design to recycling. MSCoRe addresses this by focusing not just on domain knowledge, but on reasoning across the procedural stages of that domain.

## 2.3 MULTI-STEP REASONING IN LLM AGENTS

The challenge of enabling LLMs to solve complex, multi-step problems has been a major driver of research. The introduction of Chain-of-Thought (CoT) prompting (Wei et al., 2022) was a seminal step, demonstrating that eliciting intermediate reasoning steps significantly improves performance on complex tasks. Self-Consistency (Wang et al., 2022a) improves robustness by sampling multiple reasoning paths and selecting the most consistent answer. More structured approaches like Tree of Thoughts (ToT) Yao et al. (2023) and Graph of Thoughts (GoT) (Besta et al., 2024) allow models to explore and self-evaluate diverse reasoning pathways, enabling them to handle problems that require planning and exploration. However, a corresponding gap exists in evaluation: many benchmarks testing these abilities rely on abstract mathematical problems, or game-playing scenarios. While useful, they may not reflect the constraints and dependencies of real-world industrial processes. GAIA (Mialon et al., 2023) has begun to address complex, multi-step tasks requiring tool use, but still maintains a generalist focus. MSCoRe measures a model's ability to solve problems where success is determined not by isolated logic, but by a holistic understanding of the interplay between different operational stages.

## 3 DATA

### 3.1 DATASET OVERVIEW

**Data Fotmulation** Every QA instance in MSCoRe follows the Alpaca instruction-tuning format, where each instance $d_i = (I_i, X_i, O_i)$ comprises three primary fields: *instruction* $(I_i)$, *input* $(X_i)$, and *output* $(O_i)$. The *instruction* field contains the question that guides the model's response generation. The *output* field provides the target response that serves as the ground truth for training and evaluation. Our task formulation does not utilize *input* component, thus $X_i = \emptyset$ for all instances in MSCoRe. The format without additional input allows the model to generate answers directly from instructions. This distinguishes our approach from context-dependent reasoning tasks and emphasizes the evaluation of reasoning based on models' parametric knowledge.

Table 1: Distribution of data across each dataset at Simple, Medium, and Complex levels.

| Domain | Simple | Medium | Complex | Total |
|---|---|---|---|---|
| Automotive | 56708 | 6034 | 6045 | 68787 |
| Pharmaceutical | 25032 | 2005 | 2004 | 29041 |
| E-Commerce | 17731 | – | 2005 | 19736 |
| Auto-Energy | – | 9132 | – | 9132 |

**Data Distribution** MSCoRe encompasses four critical industrial value chains: Automotive, Pharmaceutical, E-Commerce, and Automotive Energy. The specific data volume and distribution are shown in Figure 2. Detailed descriptions and examples are listed in the Appendix B.1.

**Task Definition** To enable a more systematic and fine-grained evaluation of model performance, we categorize the dataset into three difficulty levels: **Simple**, **Medium**, and **Complex**, based on the coverage of work stages.

- **Simple Tasks** focus on single-stage optimization within individual stage. For instance, considering only the fuel economy of a vehicle from the perspective of automotive design.

- **Medium Tasks** involve coordinating between two or more stages. An exemplar task requires coordinating vehicle design and manufacturing processes to optimize fuel efficiency, necessitating cross-functional understanding and multi-stage optimization.

- **Complex Tasks** demand holistic integration across all value chain stages. Tasks such as integrating design, manufacturing, supply chain, and recycling for electric vehicle optimization require comprehensive reasoning and the ability to balance competing objectives across the entire product lifecycle.

From basic single-stage reasoning to complex full-chain optimization and decision-making, the hierarchical difficulty structure ensures effectively discriminate between models with varying capabilities in different domain understanding. Table 1 summarizes the distribution of instances across domains and difficulty levels. The E-Commerce value chain comprises four stages, hence optimization tasks are defined solely at single-stage tasks (Simple) and full-chain tasks (Complex). The Automotive Energy chain mainly focuses on collaborative reasoning between vehicle domain and energy domain, thus all data are multi-stage optimization tasks (Medium).

### 3.2 DATA CONSTRUCTION

**Dynamic Sampling** To ensure comprehensive coverage across all stages within the domain and guarantee data professionalism, we manually collected a small volume of high-quality seed data as learning examples. However, repeated sampling is adverse to data diversity. So, we abandon uniform random sampling, and adopt a dynamic sampling strategy, defining a linearly decreasing sampling probability distribution for learning example sampling. In the formula shown in Figure 3 **(a)**, as the index $i$ of the generated data increases, the sampling probability $P_s$ for the seed data decrease linearly, while the sampling probability $P_g$ for new data gradually increase. $N$ denotes the predefined volume of dataset, and $\gamma$ is a hyperparameter that controls the slope of the distribution. $T$ denotes the number of examples learned during question-answer generation. This effectively avoids data subject imbalance and ensures the diversity of the dataset.

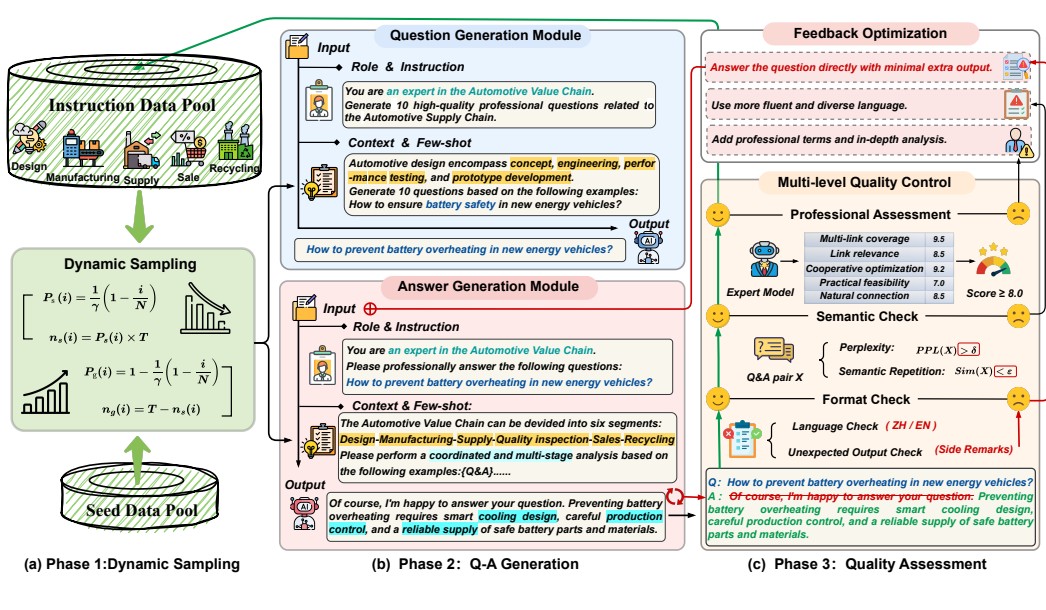

Figure 3: The framework of data construction: **(a)** Dynamic Sampling from Seed Data Pool and Instruction Data Pool. **(b)** Domain-specific Q&A instances generation based on refined prompting and high-quality learning examples. **(c)** Quality Assessment through multi-level evaluation including Format Check, Semantic Check, and Professional Assessment. Optimize data generation through Feedback Optimization strategy.

**Question-Answer Generation** The core of our data generation relies on a two-module architecture guided by refined prompt engineering, leveraging in-context learning (Dong et al., 2022) to elicit expert-level, multi-stage reasoning from the LLM agents.(As shown in Figure 3 **(b)**.)

- **Question Generation Module**. This module leverages the powerful generative capabilities of LLMs to construct specialized questions within a given domain. Refined prompts incorporate specific roles (e.g.,You are an expert in the automotive value chain), task descriptions and explicit requirements (e.g., Generate 10 high-quality professional questions...). To further anchor the generated content, prompts also include contextual information and few-shot examples. These examples provide the model with relevant background knowledge and stylistic models to guide its generation.

- **Answer Generation Module**. This module takes the questions generated in the preceding step as input, outputting optimized solutions that meet requirements for single-stage, multi-stage, or full-chain optimization. Beyond defining expert roles, it supplements contextual information with relevant value chain collaboration details (e.g., Design-Manufacturing-Supply Chain-Quality Inspection-Sales-Recycling). The model learn to perform collaborative multi-stage analysis based on the provided few-shot examples, ultimately delivering optimal solutions. For instance, when addressing battery overheating prevention, the model should integrate solutions from design, manufacturing, and supply chain stages rather than providing isolated single-stage answers.The detailed prompt template shown in Appendix B.2.

**Quality Assessment** To ensure the quality and utility of the data, we implement multi-level quality control measures and incorporate feedback optimization strategies to enhance the efficiency of data generation.(As shown in Figure 3 **(c)**.)

- **Format Check**: The first layer of rule-based filtering mechanisms involves standardizing data processing. This includes language consistency checks, as all our data is in Chinese format; and unusual output detection to prevent unexpected responses (e.g., "Of course, I'm happy to answer your question.").

- **Semantic Checks**: To ensure linguistic quality and novelty, we apply two programmatic checks. First, we filter out data pairs with high perplexity $(PPL(d_i) > \delta)$ to remove overly

simplistic or generic content. We also enforce a semantic similarity threshold ($Sim(d_i)$) < $\epsilon$) to eliminate repeated or similar data, thereby ensuring data diversity.

- **Professional Assessment**: We utilize a powerful adjudicator model to score each Q&A pair based on a set of domain-specific metrics. These include multi-link coverage, link relevance, and cooperative optimization to evaluate the quality of the multi-stage reasoning, as well as practical feasibility and natural connection to assess real-world applicability. Q&A Pairs failing to meet a predefined quality threshold (e.g., $Score \leq 8.0$) are discarded. The detailed assessment prompt is shown in Appendix B.2.

**Feedback Optimization** Our pipeline operates within a closed-loop system. During quality control, any data failing to meet requirements is fed back to the Answer Generation Module according to error type, incorporating corresponding feedback for a finite number of iterative optimizations. For instance, upon detecting unexpected outputs in responses, we explicitly introduce the instruction 'Answer the question directly with minimal extra output' to enhance conciseness and data purity in subsequent generation processes. This iterative feedback optimization mechanism achieves continuous improvement in both data quality and generation efficiency.

## 4 EXPERIMENTS

### 4.1 SETUP

We employed the GLM4 model (GLM et al., 2024) as our primary question-answering generation framework, GPT-4 served as the expert evaluator in our collaborative assessment pipeline. The following hyperparameters were empirically determined: the number of few-shot exemplars $T$ was set to 10 to provide sufficient contextual guidance while maintaining computational tractability. The collaboration weight parameter $\gamma$ was fixed at 1.0 to ensure balanced contribution from all evaluation components. For semantic validation, we established a perplexity threshold $\delta = 16$ and a similarity threshold $\epsilon = 0.9$ to maintain response coherence and relevance. The expert assessment score threshold was configured at 8.5 points on a 10-point scale to ensure high-quality output. To prevent excessive computational overhead, the maximum number of iterative refinement cycles was constrained to 2.

Table 2: LLMs performance comparison (ROUGE scores) across 4 Domains and 3 difficulty levels. darkblue indicates the highest score in each column, lightblue indicates the second highest score. A-E = Automotive Energy, Avg. = Average, Sim. = Simple, Med. = Medium, Com. = Complex.

| Model | Automotive | | | Pharmaceutical | | | E-Commerce | | A-E | Avg. |
|---|---|---|---|---|---|---|---|---|---|---|
| | Sim. | Med. | Com. | Sim. | Med. | Com. | Sim. | Com. | – | – |
| **Open-Source LLMs (Small)** | | | | | | | | | | |
| Qwen2.5-1.5B | 14.76 | 12.75 | 10.75 | 10.46 | 11.51 | 11.37 | 21.39 | 11.00 | 12.75 | 12.97 |
| Llama3.2-3B | 6.34 | 4.70 | 9.19 | 9.27 | 4.76 | 4.49 | 5.63 | 4.94 | 4.70 | 6.00 |
| Bloomz-3B | 18.55 | 19.47 | 19.66 | 16.68 | 18.36 | 17.83 | 11.53 | 16.77 | 19.47 | 17.59 |
| Qwen2.5-3B | 36.20 | 30.90 | 31.04 | 31.37 | 31.53 | 30.38 | 36.64 | 32.27 | 30.90 | 32.36 |
| **Open-Source LLMs (Medium)** | | | | | | | | | | |
| Yi-1.5-6B | 28.81 | 23.09 | 26.72 | 22.12 | 26.82 | 25.46 | 27.85 | 24.45 | 23.09 | 25.38 |
| Qwen2-7B | 42.37 | 34.98 | 36.07 | 33.27 | 35.10 | 33.12 | 43.15 | 35.08 | 34.98 | 36.46 |
| Qwen2.5-7B | 31.71 | 27.64 | 29.17 | 28.99 | 28.45 | 27.51 | 33.50 | 29.14 | 27.64 | 29.31 |
| DeepSeek-R1-7B | 46.28 | 40.82 | 43.55 | 41.81 | 40.69 | 39.28 | 45.71 | 41.82 | 40.82 | 42.18 |
| GLM4-9B | 34.18 | 25.90 | 37.74 | 26.90 | 27.12 | 26.00 | 23.79 | 26.15 | 25.90 | 28.19 |
| Qwen2.5-14B | 30.94 | 25.59 | 32.28 | 27.03 | 27.23 | 26.10 | 31.01 | 27.59 | 25.59 | 28.15 |
| Phi4-14B | 44.48 | 37.93 | 17.97 | 33.26 | 37.29 | 34.85 | 51.94 | 34.23 | 37.93 | 36.65 |
| DeepSeek-R1-14B | 46.93 | 40.78 | 40.47 | 40.71 | 39.48 | 38.29 | 49.03 | 40.10 | 40.78 | 41.84 |
| **Closed-Source LLMs** | | | | | | | | | | |
| Claude3.5-Haiku | 43.93 | 36.58 | 38.55 | 39.21 | 35.00 | 33.71 | 43.18 | 34.28 | 36.58 | 37.89 |
| GPT-3.5-Turbo | 44.28 | 38.38 | 41.29 | 35.18 | 36.73 | 34.69 | 43.13 | 35.61 | 38.38 | 38.63 |
| GPT-4o | 48.78 | 43.21 | 45.42 | 43.83 | 41.33 | 40.92 | 50.21 | 41.29 | 43.21 | 44.24 |

To quantify the multi-stage reasoning capabilities of various LLMs, we conducted an extensive evaluation on the MSCoRe. We benchmarked 15 prominent LLMs, including 12 open-source models ranging from 1.5B to 14B parameters: Qwen2, DeepSeek-R1, Phi4 (Abdin et al., 2024), Llama3.2 (Glória et al., 2024), Bloomz (Workshop et al., 2022), and Yi (Young et al., 2024), as well as 3 closed-source models: Claue-3.5-Hiaku, GPT-3.5-Turbo, and GPT-4o. The performances are measured using ROUGE scores across four domains and difficulty levels.

## 4.2 COLLABORATIVE EFFECTIVENESS ASSESSMENT

The results, presented in Table 2, reveal two primary findings. First, while the state-of-the-art closed-source model, GPT-4o achieves the highest overall performance, the leading open-source models have become remarkably competitive. DeepSeek-R1-7B and DeepSeek-R1-14B outperform other major proprietary models, indicating a narrowing gap at the frontier of reasoning capabilities. Second, a universal and significant degradation in performance as task complexity increases. This consistent trend across most models underscores that multi-stage collaborative reasoning remains a formidable challenge, validating our benchmark's effectiveness to measure this complex capability. Furthermore, the result highlights a clear correlation between model scale and performance, as well as performance disparities across different domains. Models generally scored higher in the Automotive and E-Commerce sectors compared to the more specialized Pharmaceutical domain, likely reflecting variances in their pre-training data. These results demonstrate the utility of MSCoRe in diagnosing the multi-stage collaborative reasoning abilities of LLM agents and pinpointing key areas for future improvement. Further detailed experimental data is provided in the Appendix C.2.

## 4.3 FURTHER ANALYSIS

**Robustness Analysis**

To move beyond absolute performance and specifically evaluate a model's stability in the face of increasing task complexity, we introduce a ratio-based robustness metric. This metric is defined as the ratio of a model's ROUGE score on **Complex** tasks to its score on **Simple** tasks within the same domain. A ratio approaching 1.0 indicates the model possesses exceptional robustness, signifying that its performance remains stable with minimal degradation when transitioning from simple single-stage problems to complex multi-stage reasoning challenges.

$$\text{Robustness Ratio} = \frac{\text{ROUGE Score (Com.)}}{\text{ROUGE Score (Sim.)}}$$

The result reveals model-specific sensitivities to different domains. Phi4-14B, exhibits extreme brittleness in the Automotive domain with a ratio of only 0.40. However, in the Pharmaceutical domain, its ratio soars to 1.05. This volatility indicates that a model's reasoning stability is not an intrinsic general property but is highly contingent on its familiarity with a specific domain's relational knowledge. This phenomenon does not necessarily imply superior reasoning on harder tasks. A more plausible explanation is a stylistic artifact related to the ROUGE metric itself: **Complex** tasks often require longer, more comprehensive answers involving multiple stages. Models that are inherently more verbose may achieve higher lexical overlap on these tasks despite imperfect underlying logic.

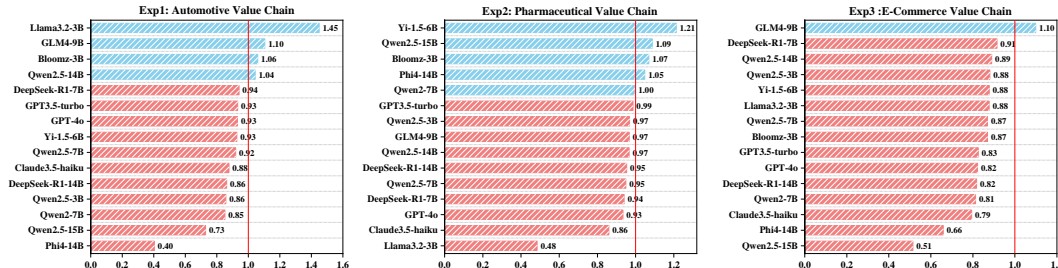

Figure 4: Robustness evaluation across three value chain domains: performance ratio of Complex-to-Simple tasks for various LLMs on Automotive, Pharmaceutical , and E-Commerce value chain. Higher Robustness Ratios (blue bars) indicate better robustness, while lower Robustness Ratios (red bars) suggest performance degradation on challenging tasks.

**Few-Shot-Learning Analysis**

To investigate the impact of in-context learning on multi-stage reasoning, we conducted a comparative analysis between zero-shot (K=0) and one-shot (K=1) settings. Due to resource constraints, our analysis focused on four models: Bloomz-3B, Qwen2.5-7B, DeepSeek-R1-14B, and GPT-3.5-Turbo. The results are presented in the Table 3. For Bloomz-3B and Qwen2.5-7B, the one-shot prompt provided a noticeable performance uplift. This suggests that for these models, the in-context example successfully demonstrates a complex reasoning pattern that is difficult to elicit in a zero-shot setting. In stark contrast, the highly capable models with strong zero-shot performance, DeepSeek-R1-14B and GPT-3.5-Turbo, exhibited a consistent and significant degradation in performance when provided with a one-shot example. This counter-intuitive result points to the high prompt sensitivity of these advanced models. We hypothesize that models with robust internal strategies for these tasks may find a single in-context example misaligned with their inherent reasoning process, thereby constraining them to a suboptimal solution path. This highlights a key challenge in multi-stage collaborative reasoning: in-context learning is not universally beneficial and may even hinder strong models.

Table 3: Model performance for 0-shot and 1-shot learning

| Model | k-shot | Sim. | Med. | Com. |
|-------|--------|------|------|------|
| Bloomz-3B | 0 | 19.72 | 19.64 | 19.34 |
|  | 1 | 25.84 | 20.16 | 20.53 |
| Qwen2.5-7B | 0 | 30.97 | 26.73 | 14.8 |
|  | 1 | 34.42 | 29.44 | 28.78 |
| DeepSeek-14B | 0 | 46.67 | 41.41 | 40.28 |
|  | 1 | 42.75 | 35.69 | 31.78 |
| GPT-3.5-Turbo | 0 | 43.53 | 37.75 | 38.86 |
|  | 1 | 37.82 | 35.68 | 33.72 |

.

**Noise Resistance**

To further evaluate the model's multi-stage collaborative optimization capability on noisy data, we introduced noise into 2,000 data points sampled in E-Commerce Value Chain Complex tasks. We selected three representative models, ChatGLM3-6B, DeepSeek-R1-7B, and Baichuan2-13B for experimentation. We establish three common types of noise:Format Corruption, Syntactic Confusion, and Character Omission. Different noise rates were also set for comparison. The results are shown in the Figure 5. Detailed explanations of noises and implementation methods are provided in the Appendix B.3.

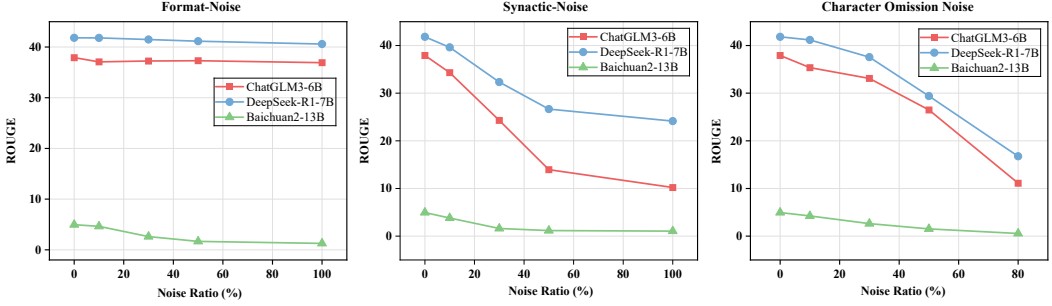

Figure 5: ROUGE scores for the model under three different noise types and varying noise rates

Under the influence of three types of noise, the robustness of the three models exhibited significant differences. Firstly, formatting noise had a relatively minor impact on the models, all three demonstrated a gradual decline in Rouge scores as the noise proportion increased, indicating a high

tolerance for structural perturbations. Secondly, syntactic confusion exerted a greater impact on performance, with Rouge scores declining markedly under high noise ratios. This indicates deficiencies in capturing syntactic sequences and dependency relationships. Finally, character omissions caused the most severe performance degradation, highlighting the central role of semantic integrity in generation quality. Moreover, Baichuan 2-13B demonstrated the strongest overall robustness, followed by DeepSeek-R1-7B, while ChatGLM3-6B performed worst under high-noise conditions. This indicates that larger models exhibit greater fault tolerance when confronting multiple noise types.

**Error Analysis**

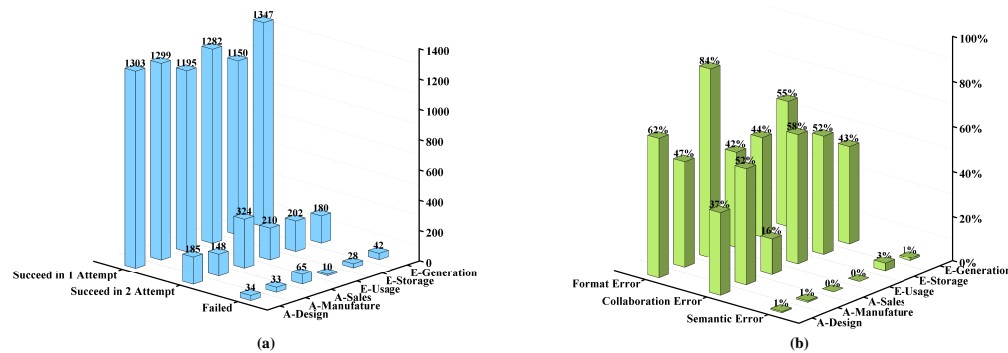

Figure 6: Success counts (a) and error-type proportions (b) in automated data generation across the Automotive Energy value chain.

To further investigate the reliability of automated data generation, we analyzed patterns of successful and failed data generation across different segments of the automotive-energy value chain. As shown in Figure 6 **(a)**, the majority of data instances were successfully generated within one or two attempts, while a relatively small proportion persisted in failing despite multiple retries. To delve deeper into these failure cases, Figure 6 **(b)** categorizes error types statistically. Semantic errors account for the smallest proportion of failures, with the primary failure types being insufficient coordination and formatting errors. This may stem from the persistent difficulty in maintaining consistency across multi-stage reasoning, coupled with inadequate adherence to target formats or constraints.

## 5 CONCLUSION

We propose MSCoRe, a novel benchmarking framework for systematically evaluating the multi-stage collaborative reasoning capabilities of LLM agents across four key industrial value chains: automotive, pharmaceutical, e-commerce, and automotive energy. Through comprehensive experiments on 15 state-of-the-art models, we observe that while commercially advanced LLMs demonstrate overall strongest performance, they still face significant challenges in full-chain reasoning tasks and exhibit pronounced fragility under noisy or unstructured inputs. These findings indicate current large models remain inadequate for multi-stage collaborative reasoning tasks. MSCoRe establishes a data foundation for evaluating model multi-stage collaborative capabilities, laying crucial groundwork for enhancing future models' reasoning capacity, robustness, and practical application value within complex processes.

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

## A    LLM USAGE

Large Language Models (LLMs) were used to aid in the writing and polishing of the manuscript. Specifically, we used an LLM to assist in refining the language, improving readability, and ensuring clarity in various sections of the paper. The model helped with tasks such as sentence rephrasing, grammar checking, and enhancing the overall flow of the text. It is important to note that the LLM was not involved in the ideation, research methodology, or experimental design. All research concepts, ideas, and analysis were developed and conducted by the authors. The contributions of the LLM were solely focused on improving the linguistic quality of the paper, with no involvement in the scientific content or data analysis.

The authors take full responsibility for the content of the manuscript, including any text generated or polished by the LLM. We have ensured that the LLM-generated text adheres to ethical guidelines and does not contribute to plagiarism or scientific misconduct.

## B    APPENDIX

### B.1    DATASET DESCRIPTION

- **Automotive Value Chain**: Encompassing the entire lifecycle process from vehicle design, production, supply chain management, and quality inspection to sales and recycling. Through coordinated efforts across all stages, it creates value while driving industrial competitiveness and sustainable development. The detailed description and example of each stage is shown in Table 4.

- **Pharmaceutical Value Chain**: Encompassing the entire process from drug research and development, raw material procurement, drug manufacturing, supply chain management, to sales and distribution. Through coordination across all stages, it drives pharmaceutical innovation, quality assurance, and market accessibility. The detailed description and example of each stage is shown in Table 4.

- **E-Commerce Value Chain**: Encompassing the entire process from product procurement, warehousing and logistics, platform operations, to user experience. It involves the coordinated optimization of product supply, delivery efficiency, platform traffic, and user satisfaction across all stages. The detailed description and example of each stage is shown in Table 5.

- **Automotive Energy Chain**: Encompassing the entire process spanning vehicle design, production, sales, and energy systems (usage, energy storage, power generation) in synergy. By integrating vehicle manufacturing with energy supply, it achieves efficient energy utilization, reduces costs, and promotes the sustainable development of new energy vehicles. The detailed description and example of each stage is shown in Table 5.

### B.2    PROMPT TEMPLATE

- Question generation prompt template of each stage in the Automotive Value Chain are shown in Table 11 and Table 12.

- Answer generation template of each stage in the Automotive Value Chain are shown in Table 13 and Table 14

- Professional assessment prompt template is shown in Figure 8.

### B.3    NOISE ANALYSIS

- **Format Corruption**, refers to textual anomalies in layout or structure, such as disordered paragraphs, resulting in non-standardized presentation of information. we introduce noise by adding line breaks.

- **Syntactic Confusion**, Refers to the disruption of the original grammatical structure and semantic coherence within a sentence, resulting from the rearrangement of words or phrases. Achieved by altering word order.

- **Character Omission**, refers to the omission or ellipsis of certain key information within a text, resulting in incomplete meaning or unclear expression. Achieved by randomly removing words.

Table 4: The Automotive Value Chain comprises 6 stages, while the Pharmaceutical Value Chain consists of 5 stages, along with detailed descriptions and illustrative examples for each stage.

| Domain | Stage | Description | Example |
|---|---|---|---|
| **Automotive** | Design | Conducting design planning and performance optimization for complete vehicles and components in accordance with market demands and regulatory requirements. | How can weight be reduced in vehicle body design to enhance fuel efficiency? |
| | Manufacturing | Transforming designs into high-quality finished vehicles through automation and lean manufacturing. | How can we reduce energy consumption and scrap rates on the assembly line? |
| | Supply Chain | Integrate the procurement of raw materials and components, logistics, and inventory management to ensure stable production. | How can we ensure the timely delivery of critical components to avoid production delays? |
| | Quality Inspection | Rigorous testing and inspection of components and complete vehicles to ensure compliance with safety and quality standards. | How can one swiftly detect potential safety hazards in electric vehicle batteries? |
| | Sales | Drive product adoption and facilitate transactions through channel development, marketing, and customer service. | How can the user purchasing experience for new energy vehicles be enhanced? |
| | Recycle | Dismantling end-of-life vehicles for material recovery and remanufacturing reduces resource consumption and environmental impact. | How can batteries and metallic materials be efficiently recovered from end-of-life vehicles? |
| **Pharmaceutical** | Drug Development | The efficacy and safety of new drugs are identified and validated through fundamental research and clinical trials. | How can we improve the success of clinical trials for new drugs? |
| | Material Procurement | Select and procure high-quality, compliant medicinal raw materials to meet production requirements. | How can we ensure the stability of raw material supply and consistency in quality? |
| | Manufacturing | Process raw materials into marketable pharmaceutical products in accordance with stringent manufacturing processes and GMP standards. | How can production efficiency be enhanced whilst maintaining pharmaceutical quality? |
| | Supply Chain | Plan and optimize the storage, transportation and distribution of pharmaceuticals to ensure safe, compliant and timely delivery. | How to safely transport vaccines in low-temperature environments? |
| | Sales & Distribution | Promoting pharmaceutical products to healthcare institutions and retail outlets to enhance market coverage and accessibility. | How can we increase the prescription usage rate of new drugs in hospitals? |

Table 5: The E-Commerce Value Chain comprises 4 stages, while the Automotive Energy Chain consists of 5 stages, along with detailed descriptions and illustrative examples for each stage.

| Domain | Stage | Description | Example |
|---|---|---|---|
| **E-Commerce** | Product Procurement | Select and procure high-quality, competitively priced goods to meet customer requirements. | How can one quickly assess the market popularity and sales potential of a particular product category? |
| | Warehousing & Logistics | Efficiently manage inventory and optimize delivery processes to ensure goods are delivered to customers promptly and accurately. | How can we reduce warehousing costs and improve dispatch speed? |
| | Platform Operations | Enhance platform traffic and conversion rates through data analysis, marketing, and event planning. | How can big data be leveraged to deliver precise product recommendations to users? |
| | User Experience | Optimize the interface, customer service and after-sales support to enhance user satisfaction and repeat purchase rates. | How can we reduce user drop-off rates during the checkout process? |
| **Automotive Energy** | Auto-Design | Optimize the vehicle structure and powertrain system according to the characteristics of new energy to enhance range and safety. | How can energy losses in electric vehicles be minimized during the design phase? |
| | Auto-Manufacturing | Employ intelligent manufacturing and quality control technologies to efficiently produce new energy vehicles. | How can consistency be ensured for power batteries during the production process? |
| | Auto-Sales | Promoting market acceptance of new energy vehicle models through multi-channel marketing and service initiatives. | How can consumer confidence in electric vehicle range be enhanced? |
| | Energy-Usage | Develop and optimize the charging network to enhance charging convenience and efficiency. | How can we reduce queuing times for electric vehicle charging during peak periods? |
| | Energy-Storage | Utilizing batteries or energy storage systems to balance grid supply and demand and support efficient energy utilization. | How can the service life of energy storage equipment at charging stations be extended? |
| | Energy-Generation | Developing clean energy generation to provide a sustainable power source for electric vehicles. | How can the power generation efficiency of photovoltaic power stations be enhanced during rainy weather? |

## C EXPERIMENTS SND ANALYSIS

### C.1 HUMAN-AI DISCRIMINATION TEST

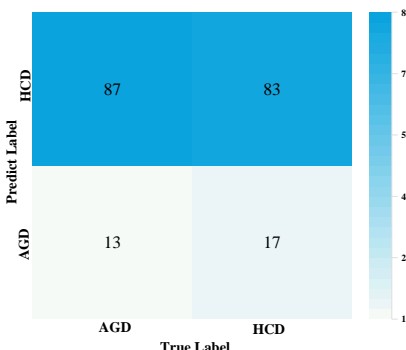

Figure 7: Confusion matrix for the Human-AI Discrimination Test showing expert classification results between automatically generated data (AGD) and human-created data (HCD).

To further validate the data quality of MSCoRe, we designed a Human-AI Discrimination Test, wherein ten domain experts distinguished whether test data belonged to automatically generated data or manually created data. We presented the experts with a mixed dataset, requesting they classify each data as either AGD (Automatically Generated Data) or HCD (Human Created Data). The experimental results are summarized in the confusion matrix shown in Figure 7. A significant majority, 87.0% of the AGD samples, were incorrectly classified as human-created. This high misclassification rate strongly suggests that our generated data exhibits characteristics of human-level quality and sophistication, making it difficult for experts to distinguish from authentic human work.

### C.2 EXPERIMENTAL RESULTS

BLEU and ROUGE scores for each model across four datasets.Table 6, Table 7, Table 8, Table 9, and Table 10.

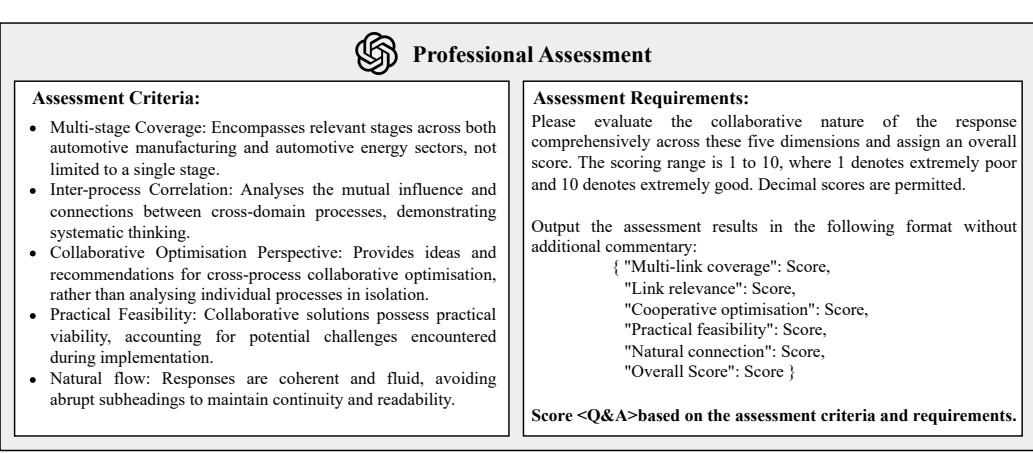

Figure 8: The Professional Assessment prompt template.

Table 6: LLMs performance comparison (BLEU and ROUGE scores) in six stages within Automotive Energy Chain. darkblue indicates the highest score in each column, lightblue indicates the second highest score.

| Model | Stage-1 | | Stage-2 | | Stage-3 | | Stage-4 | | Stage-5 | | Stage-6 | | Avg. | |
|---|---|---|---|---|---|---|---|---|---|---|---|---|---|---|
| | B | R | B | R | B | R | B | R | B | R | B | R | B | R |
| **Open-Source LLMs (Small)** | | | | | | | | | | | | | | |
| Qwen2.5-1.5B | 3.83 | 12.78 | 4.25 | 13.70 | 4.33 | 13.27 | 2.68 | 9.24 | 3.78 | 12.15 | 5.15 | 15.33 | 4.00 | 12.75 |
| Llama3.2-3B | 0.79 | 4.97 | 0.84 | 4.88 | 0.99 | 5.04 | 0.68 | 4.22 | 0.74 | 4.60 | 0.67 | 4.50 | 0.79 | 4.70 |
| Bloomz-3B | 3.31 | 19.67 | 3.65 | 20.41 | 3.97 | 18.74 | 3.70 | 19.69 | 3.57 | 19.54 | 3.08 | 18.78 | 3.55 | 19.47 |
| Qwen2.5-3B | 10.69 | 31.64 | 10.60 | 32.00 | 9.56 | 29.67 | 9.80 | 30.43 | 10.39 | 30.79 | 10.97 | 30.86 | 10.34 | 30.90 |
| **Open-Source LLMs (Medium)** | | | | | | | | | | | | | | |
| Yi-1.5-6B | 6.15 | 22.97 | 6.07 | 23.94 | 5.85 | 22.06 | 5.81 | 22.85 | 6.86 | 24.03 | 6.09 | 22.67 | 6.14 | 23.09 |
| Qwen2-7B | 12.41 | 35.75 | 11.78 | 36.16 | 13.25 | 33.78 | 14.37 | 34.31 | 11.06 | 34.30 | 11.62 | 35.58 | 12.42 | 34.98 |
| Qwen2.5-7B | 8.13 | 29.17 | 7.97 | 28.73 | 6.13 | 24.99 | 6.69 | 26.54 | 7.78 | 27.95 | 8.55 | 28.44 | 7.54 | 27.64 |
| DeepSeek-R1-7B | 20.49 | 41.95 | 20.56 | 41.82 | 18.61 | 39.64 | 17.31 | 39.28 | 19.62 | 40.44 | 26.27 | 41.81 | 20.48 | 40.82 |
| GLM4-9B | 11.83 | 28.31 | 9.91 | 27.63 | 8.35 | 24.34 | 9.68 | 24.32 | 8.89 | 25.26 | 9.13 | 25.56 | 9.63 | 25.90 |
| Qwen2.5-14B | 7.49 | 26.84 | 7.36 | 26.44 | 6.74 | 24.51 | 6.70 | 24.48 | 7.34 | 25.93 | 7.23 | 25.36 | 7.14 | 25.59 |
| Phi4-14B | 21.48 | 39.43 | 20.28 | 37.85 | 20.15 | 36.67 | 20.48 | 37.87 | 20.28 | 37.62 | 21.32 | 38.13 | 20.67 | 37.93 |
| DeepSeek-R1-14B | 19.84 | 42.79 | 18.27 | 42.41 | 16.29 | 38.78 | 17.25 | 39.17 | 17.61 | 40.49 | 18.12 | 41.06 | 17.90 | 40.78 |
| **Closed-Source LLMs** | | | | | | | | | | | | | | |
| Claude-3.5-Haiku | 10.08 | 35.75 | 17.57 | 46.09 | 10.02 | 34.06 | 10.86 | 33.18 | 10.56 | 33.47 | 9.60 | 36.93 | 11.45 | 36.58 |
| GPT-3.5-Turbo | 12.58 | 38.04 | 10.31 | 44.68 | 15.39 | 36.71 | 14.03 | 36.03 | 13.85 | 36.64 | 11.74 | 38.15 | 12.98 | 38.38 |
| GPT-4o | 23.27 | 44.65 | 22.93 | 49.02 | 21.82 | 40.85 | 20.33 | 40.09 | 20.38 | 40.87 | 22.72 | 43.80 | 21.91 | 43.21 |

Table 7: LLMs performance comparison (BLEU and ROUGE scores) in six stages' Simple tasks within Automotive Value Chain. darkblue indicates the highest score in each column, lightblue indicates the second highest score.

| Model | Stage-1 | | Stage-2 | | Stage-3 | | Stage-4 | | Stage-5 | | Stage-6 | | Avg. | |
|---|---|---|---|---|---|---|---|---|---|---|---|---|---|---|
| | B | R | B | R | B | R | B | R | B | R | B | R | B | R |
| **Open-Source LLMs (Small)** | | | | | | | | | | | | | | |
| Qwen2.5-1.5B | 3.99 | 15.36 | 3.62 | 13.38 | 4.50 | 16.33 | 4.14 | 15.07 | 3.37 | 12.40 | 4.43 | 15.99 | 4.01 | 14.76 |
| Llama3.2-3B | 0.82 | 5.95 | 0.93 | 6.34 | 1.05 | 5.93 | 1.17 | 7.25 | 0.83 | 5.85 | 0.71 | 5.74 | 0.92 | 6.34 |
| Bloomz-3B | 2.63 | 19.72 | 2.94 | 19.61 | 2.21 | 15.97 | 2.80 | 20.25 | 2.16 | 16.73 | 3.00 | 19.03 | 2.62 | 18.55 |
| Qwen2.5-3B | 11.41 | 35.76 | 11.67 | 36.75 | 11.19 | 35.88 | 11.67 | 36.06 | 11.80 | 36.55 | 11.74 | 36.18 | 11.58 | 36.20 |
| **Open-Source LLMs (Medium)** | | | | | | | | | | | | | | |
| Yi-1.5-6B | 6.52 | 27.14 | 7.41 | 29.64 | 7.51 | 29.91 | 7.25 | 28.94 | 6.37 | 28.04 | 8.16 | 29.20 | 7.20 | 28.81 |
| Qwen2-7B | 10.72 | 41.41 | 10.30 | 42.88 | 10.32 | 42.13 | 10.33 | 42.89 | 10.83 | 42.85 | 11.73 | 42.07 | 10.71 | 42.37 |
| Qwen2.5-7B | 8.36 | 30.97 | 8.73 | 32.07 | 8.93 | 32.35 | 8.50 | 31.51 | 8.97 | 32.32 | 8.54 | 31.03 | 8.67 | 31.71 |
| DeepSeek-R1-7B | 24.03 | 45.90 | 23.97 | 47.03 | 23.30 | 46.27 | 24.03 | 46.47 | 24.09 | 46.27 | 23.70 | 23.85 | 20.48 | 46.28 |
| GLM4-9B | 10.86 | 35.15 | 9.17 | 36.22 | 7.49 | 30.91 | 11.83 | 38.60 | 7.52 | 32.15 | 8.94 | 32.07 | 9.30 | 34.18 |
| Qwen2.5-14B | 7.67 | 30.24 | 7.89 | 30.77 | 8.34 | 31.52 | 7.80 | 30.34 | 8.41 | 31.66 | 8.42 | 31.09 | 8.09 | 34.18 |
| Phi4-14B | 24.81 | 42.89 | 24.63 | 44.75 | 22.97 | 43.74 | 24.49 | 45.31 | 27.57 | 46.39 | 27.62 | 43.81 | 25.35 | 44.48 |
| DeepSeek-R1-14B | 21.79 | 46.67 | 22.63 | 47.47 | 21.21 | 46.83 | 24.34 | 48.40 | 22.77 | 47.49 | 19.74 | 44.73 | 22.08 | 46.93 |
| **Closed-Source LLMs** | | | | | | | | | | | | | | |
| Claude-3.5-Haiku | 17.77 | 43.91 | 17.57 | 46.09 | 12.76 | 44.39 | 12.51 | 43.01 | 13.13 | 43.38 | 13.37 | 42.81 | 14.52 | 43.93 |
| GPT-3.5-Turbo | 10.25 | 43.53 | 10.31 | 44.68 | 10.87 | 44.26 | 9.71 | 44.33 | 11.47 | 44.70 | 12.37 | 44.19 | 10.83 | 44.28 |
| GPT-4o | 23.44 | 48.42 | 22.93 | 49.02 | 22.10 | 48.78 | 22.54 | 48.92 | 24.41 | 49.23 | 24.87 | 48.33 | 23.38 | 48.78 |

Table 8: LLMs performance comparison (BLEU and ROUGE scores) in six stages' Medium tasks within Automotive Value Chain. darkblue indicates the highest score in each column, lightblue indicates the second highest score.

| Model | Stage-1 | | Stage-2 | | Stage-3 | | Stage-4 | | Stage-5 | | Stage-6 | | Avg. | |
|---|---|---|---|---|---|---|---|---|---|---|---|---|---|---|
| | B | R | B | R | B | R | B | R | B | R | B | R | B | R |
| **Open-Source LLMs (Small)** | | | | | | | | | | | | | | |
| Qwen2.5-1.5B | 3.91 | 11.88 | 3.88 | 12.38 | 3.21 | 10.32 | 2.81 | 8.92 | 4.12 | 13.49 | 3.94 | 12.41 | 4.00 | 12.75 |
| Llama3.2-3B | 2.04 | 8.11 | 1.65 | 7.40 | 1.96 | 7.65 | 1.94 | 7.80 | 2.25 | 9.24 | 2.35 | 9.35 | 0.79 | 4.70 |
| Bloomz-3B | 4.72 | 19.64 | 4.90 | 21.24 | 4.96 | 20.47 | 4.81 | 20.98 | 4.66 | 19.74 | 5.16 | 20.64 | 3.55 | 19.47 |
| Qwen2.5-3B | 11.33 | 31.05 | 12.10 | 34.09 | 11.72 | 33.23 | 11.89 | 32.71 | 10.07 | 31.56 | 11.31 | 31.83 | 10.34 | 30.90 |
| **Open-Source LLMs (Medium)** | | | | | | | | | | | | | | |
| Yi-1.5-6B | 8.14 | 26.85 | 8.87 | 29.34 | 9.86 | 29.44 | 8.66 | 27.24 | 9.22 | 28.29 | 9.18 | 29.21 | 6.14 | 23.09 |
| Qwen2-7B | 10.95 | 35.30 | 12.52 | 38.78 | 12.46 | 38.06 | 11.66 | 36.49 | 11..97 | 35.90 | 10.74 | 37.09 | 12.42 | 34.98 |
| Qwen2.5-7B | 8.19 | 26.73 | 8.71 | 28.78 | 8.50 | 27.54 | 8.45 | 26.98 | 7.44 | 26.28 | 9.25 | 28.32 | 7.54 | 27.64 |
| DeepSeek-R1-7B | 21.80 | 41.26 | 21.13 | 44.46 | 21.76 | 44.34 | 22.46 | 43.28 | 20.98 | 43.05 | 21.28 | 41.01 | 20.48 | 40.82 |
| GLM4-9B | 13.90 | 33.94 | 19.04 | 40.51 | 17.74 | 38.90 | 15.01 | 36.13 | 14.07 | 33.60 | 14.38 | 34.86 | 9.63 | 25.90 |
| Qwen2.5-14B | 10.45 | 29.71 | 10.46 | 31.46 | 10.51 | 31.37 | 10.65 | 30.30 | 9.99 | 30.86 | 10.11 | 29.86 | 7.14 | 25.59 |
| Phi4-14B | 3.01 | 13.11 | 4.22 | 17.40 | 3.06 | 14.21 | 20.48 | 37.87 | 20.28 | 37.62 | 21.32 | 38.13 | 20.67 | 37.93 |
| DeepSeek-R1-14B | 19.84 | 42.79 | 18.27 | 42.41 | 16.29 | 38.78 | 17.25 | 39.17 | 17.61 | 40.49 | 18.12 | 41.06 | 17.90 | 40.78 |
| **Closed-Source LLMs** | | | | | | | | | | | | | | |
| Claude-3.5-Haiku | 10.08 | 35.75 | 17.57 | 46.09 | 10.02 | 34.06 | 10.86 | 33.18 | 10.56 | 33.47 | 9.60 | 36.93 | 11.45 | 36.58 |
| GPT-3.5-Turbo | 12.58 | 38.04 | 10.31 | 44.68 | 15.39 | 36.71 | 14.03 | 36.03 | 13.85 | 36.64 | 11.74 | 38.15 | 12.98 | 38.38 |
| GPT-4o | 23.27 | 44.65 | 22.93 | 49.02 | 21.82 | 40.85 | 20.33 | 40.09 | 20.38 | 40.87 | 22.72 | 43.80 | 21.91 | 43.21 |

Table 9: LLMs performance comparison (BLEU and ROUGE scores) in six stages' Complex tasks within Automotive Value Chain. darkblue indicates the highest score in each column, lightblue indicates the second highest score.

| Model | Stage-1 | | Stage-2 | | Stage-3 | | Stage-4 | | Stage-5 | | Stage-6 | | Avg. | |
|---|---|---|---|---|---|---|---|---|---|---|---|---|---|---|
| | B | R | B | R | B | R | B | R | B | R | B | R | B | R |
| **Open-Source LLMs (Small)** | | | | | | | | | | | | | | |
| Qwen2.5-1.5B | 3.48 | 10.84 | 3.40 | 10.82 | 3.45 | 11.01 | 3.40 | 10.35 | 2.73 | 8.83 | 4.27 | 12.67 | 3.46 | 10.75 |
| Llama3.2-3B | 2.16 | 9.28 | 2.44 | 10.38 | 2.09 | 9.03 | 2.22 | 9.27 | 1.78 | 8.38 | 2.19 | 8.79 | 2.15 | 9.19 |
| Bloomz-3B | 4.42 | 19.34 | 4.55 | 20.10 | 5.04 | 20.31 | 4.55 | 19.99 | 4.31 | 18.52 | 4.72 | 19.72 | 4.60 | 19.66 |
| Qwen2.5-3B | 12.13 | 32.54 | 12.74 | 34.45 | 5.04 | 20.31 | 12.51 | 33.45 | 10.78 | 32.12 | 12.19 | 33.36 | 10.90 | 31.04 |
| **Open-Source LLMs (Medium)** | | | | | | | | | | | | | | |
| Yi-1.5-6B | 7.63 | 25.51 | 8.16 | 26.75 | 8.87 | 26.82 | 8.70 | 26.70 | 8.55 | 26.88 | 9.26 | 27.66 | 8.53 | 26.72 |
| Qwen2-7B | 10.75 | 35.10 | 11.22 | 38.06 | 11.35 | 38.06 | 11.69 | 32.07 | 9.86 | 35.52 | 10.70 | 37.60 | 10.93 | 36.07 |
| Qwen2.5-7B | 8.97 | 28.69 | 9.81 | 30.27 | 9.20 | 29.97 | 9.86 | 29.43 | 8.34 | 28.02 | 9.11 | 28.61 | 9.22 | 29.17 |
| DeepSeek-R1-7B | 21.39 | 42.28 | 22.55 | 43.89 | 22.16 | 44.88 | 22.01 | 43.22 | 21.34 | 43.93 | 21.78 | 43.07 | 21.87 | 43.55 |
| GLM4-9B | 16.10 | 36.93 | 15.88 | 38.33 | 15.33 | 38.44 | 16.35 | 37.88 | 14.38 | 36.93 | 16.80 | 37.90 | 15.81 | 37.74 |
| Qwen2.5-14B | 11.53 | 32.30 | 11.85 | 33.48 | 11.49 | 33.18 | 11.39 | 32.07 | 10.06 | 30.70 | 11.27 | 31.92 | 11.27 | 32.28 |
| Phi4-14B | 3.08 | 17.88 | 3.47 | 18.55 | 4.11 | 18.31 | 4.10 | 18.82 | 3.55 | 16.44 | 4.14 | 17.83 | 3.47 | 17.97 |
| DeepSeek-R1-14B | 16.18 | 40.26 | 16.88 | 42.42 | 15.87 | 40.99 | 18.23 | 41.75 | 15.11 | 36.92 | 15.65 | 40.45 | 16.32 | 40.47 |
| **Closed-Source LLMs** | | | | | | | | | | | | | | |
| Claude-3.5-Haiku | 12.71 | 38.54 | 13.32 | 38.69 | 14.02 | 39.78 | 13.17 | 36.46 | 12.71 | 38.14 | 13.06 | 39.66 | 13.17 | 38.55 |
| GPT-3.5-Turbo | 12.46 | 43.60 | 12.36 | 40.90 | 13.25 | 41.48 | 13.56 | 40.28 | 12.77 | 39.93 | 12.57 | 40.52 | 12.83 | 41.29 |
| GPT-4o | 20.76 | 44.60 | 20.96 | 46.56 | 20.21 | 45.95 | 21.09 | 45.26 | 20.19 | 44.61 | 20.70 | 45.55 | 20.65 | 45.42 |

Table 10: LLMs performance comparison (BLEU and ROUGE scores) in five stages' Simple Tasks within Pharmaceutical Value Chain. darkblue indicates the highest score in each column, lightblue indicates the second highest score.

| Model | Stage-1 | | Stage-2 | | Stage-3 | | Stage-4 | | Stage-5 | | Avg. | |
|---|---|---|---|---|---|---|---|---|---|---|---|---|
| | B | R | B | R | B | R | B | R | B | R | B | R |
| **Open-Source LLMs (Small)** | | | | | | | | | | | | |
| Qwen2.5-1.5B | 5.42 | 14.28 | 2.81 | 8.94 | 3.60 | 10.29 | 3.65 | 10.69 | 2.60 | 8.08 | 3.62 | 10.46 |
| Llama3.2-3B | 2.94 | 10.53 | 2.10 | 7.88 | 2.74 | 9.54 | 2.50 | 9.42 | 2.63 | 9.00 | 2.85 | 9.27 |
| Bloomz-3B | 2.67 | 17.69 | 2.65 | 16.18 | 2.64 | 17.28 | 2.53 | 16.13 | 3.32 | 16.13 | 2.76 | 16.68 |
| Qwen2.5-3B | 12.38 | 30.47 | 12.94 | 31.72 | 12.91 | 31.36 | 13.52 | 31.70 | 13.57 | 31.61 | 13.06 | 31.37 |
| **Open-Source LLMs (Medium)** | | | | | | | | | | | | |
| Yi-1.5-6B | 7.07 | 21.98 | 7.07 | 23.92 | 6.05 | 22.26 | 5.41 | 19.32 | 7.43 | 23.11 | 6.61 | 22.12 |
| Qwen2-7B | 5.30 | 32.88 | 4.46 | 33.51 | 4.35 | 34.57 | 3.72 | 32.72 | 3.55 | 32.66 | 4.28 | 33.27 |
| Qwen2.5-7B | 10.02 | 28.36 | 10.41 | 28.85 | 11.29 | 30.09 | 10.75 | 29.06 | 10.51 | 28.61 | 10.60 | 28.99 |
| DeepSeek-R1-7B | 25.29 | 41.98 | 25.54 | 41.85 | 26.21 | 41.94 | 27.36 | 42.23 | 26.96 | 41.05 | 26.27 | 41.81 |
| GLM4-9B | 6.83 | 29.45 | 3.56 | 24.22 | 5.48 | 31.01 | 3.51 | 26.97 | 2.76 | 22.85 | 4.43 | 26.90 |
| Qwen2.5-14B | 8.55 | 25.65 | 9.06 | 27.19 | 9.78 | 28.11 | 9.01 | 26.84 | 9.41 | 27.34 | 9.16 | 27.03 |
| Phi4-14B | 21.31 | 40.02 | 20.47 | 37.24 | 20.66 | 38.10 | 20.78 | 37.14 | 18.64 | 33.33 | 20.77 | 33.26 |
| DeepSeek-R1-14B | 20.42 | 42.77 | 17.26 | 39.87 | 19.52 | 43.42 | 16.03 | 39.63 | 16.87 | 37.85 | 18.02 | 40.71 |
| **Closed-Source LLMs** | | | | | | | | | | | | |
| Claude-3.5-Haiku | 9.79 | 42.88 | 6.76 | 37.69 | 7.58 | 39.46 | 4.48 | 37.65 | 7.03 | 38.36 | 7.13 | 39.21 |
| GPT-3.5-Turbo | 3.01 | 35.21 | 3.88 | 35.10 | 3.35 | 36.40 | 2.62 | 34.98 | 2.86 | 34.23 | 3.14 | 35.18 |
| GPT-4o | 17.41 | 45.80 | 14.86 | 43.44 | 13.85 | 44.64 | 14.03 | 44.08 | 12.66 | 41.18 | 14.56 | 43.83 |

Table 11: Domain-specific prompt templates for question generation of design, manufacturing, and supply chain in the automotive value chain. <Role>defines the model's expert identity. <Context>establishes domain knowledge boundaries, and <Requirements>specify output quality and format constraints.<Example>are examples sampled in Phase-1: Dynamic Sampling.

| Stage | Prompt |
|---|---|
| Design | **<Role>:** You are an automotive design expert with comprehensive expertise across all aspects of automotive design.

**<Context>:** Automotive design spans the entire value chain: concept design, engineering design, performance testing, and prototype development. <Example 1>;<Example 2>......

**<Requirements>:** Generate 10 high-quality professional questions on automotive design with these requirements:
  1. Focus on the automotive design process.
  2. Address cutting-edge challenges and current industry trends.
  3. Provide sufficient technical depth for expert-level discussion.
  4. Offer practical value for real-world challenges.
  5. Cover diverse automotive design facets.

List 10 questions directly in numbered format without additional explanations or background context. |
| Manufacturing | **<Role>:** You are an automotive manufacture expert with comprehensive expertise across all aspects of automotive manufacture.

**<Context>:**Automotive manufacturing encompassing assembly line design, manufacturing processes, and quality control.
<Example 1>;<Example 2>......

**<Requirements>:** Generate 10 high-quality professional questions on automotive manufacturing with these requirements:
  1. Focus on the automotive manufacture process.
  2. Address cutting-edge challenges and current industry trends.
  3. Provide sufficient technical depth for expert-level discussion.
  4. Offer practical value for real-world challenges.
  5. Cover diverse automotive manufacture facets.

List 10 questions directly in numbered format without additional explanations or background context. |
| Supply Chain | **<Role>:** You are an automotive supply chain expert with comprehensive expertise across all aspects of automotive supply chain.

**<Context>:** Automotive supply chain encompasses areas such as parts procurement, supplier management, and logistics planning.
<Example 1>;<Example 2>......

**<Requirements>:** Generate 10 high-quality professional questions on automotive supply chain with these requirements:
  1. Focus on the automotive supply chain process.
  2. Address cutting-edge challenges and current industry trends.
  3. Provide sufficient technical depth for expert-level discussion.
  4. Offer practical value for real-world challenges.
  5. Cover diverse automotive supply chain facets.

List 10 questions directly in numbered format without additional explanations or background context. |

Table 12: Domain-specific prompt templates for question generation of quality inspection, sales, and recycling in the automotive value chain. <Role>defines the model's expert identity. <Context>establishes domain knowledge boundaries, and <Requirements>specify output quality and format constraints.<Example>are examples sampled in Phase-1: Dynamic Sampling.

| Stage | Prompt |
|---|---|
| Quality Inspection | **<Role>:** You are an automotive quality inspection expert with comprehensive expertise across all aspects of automotive quality control.

**<Context>:** Automotive quality inspection spans whole-vehicle testing, safety assessments, and performance verification.
<Example 1>;<Example 2>......

**<Requirements>:** Generate 10 high-quality professional questions on automotive quality inspection with these requirements:

  1. Focus on the automotive quality inspection process.
  2. Address cutting-edge challenges and current industry trends.
  3. Provide sufficient technical depth for expert-level discussion.
  4. Offer practical value for real-world challenges.
  5. Cover diverse automotive quality inspection facets.

List 10 questions directly in numbered format without additional explanations or background context. |
| Sales | **<Role>:** You are an automotive sales expert with comprehensive expertise across all aspects of automotive sales.

**<Context>:** Automotive sales marketing, dealer networks, and customer service. <Example 1>;<Example 2>......

**<Requirements>:** Generate 10 high-quality professional questions on automotive sales with these requirements:

  1. Focus on the automotive sales process.
  2. Address cutting-edge challenges and current industry trends.
  3. Provide sufficient technical depth for expert-level discussion.
  4. Offer practical value for real-world challenges.
  5. Cover diverse automotive sales facets.

List 10 questions directly in numbered format without additional explanations or background context. |
| Recycle | **<Role>:** You are an automotive recycling expert with comprehensive expertise across all aspects of automotive recycling.

**<Context>:** Automotive recycling spans the processing of end-of-life vehicles, the reuse of components, and environmental protection measures. <Example 1>;<Example 2>......

**<Requirements>:** Generate 10 high-quality professional questions on automotive recycling with these requirements:

  1. Focus on the automotive recycling process.
  2. Address cutting-edge challenges and current industry trends.
  3. Provide sufficient technical depth for expert-level discussion.
  4. Offer practical value for real-world challenges.
  5. Cover diverse automotive recycling facets.

List 10 questions directly in numbered format without additional explanations or background context. |

Table 13: Domain-specific prompt templates for answer generation of design, manufacturing, and supply chain in the automotive value chain. <Role>defines the model's expert identity. <Context>establishes domain knowledge boundaries, and <Requirements>specify output quality and format constraints.<Example>are examples sampled in Phase-1: Dynamic Sampling. <Question>is generated in Phase-2: Question Generation Module.

| Stage | Prompt |
|---|---|
| Design | **<Role>:** You are a senior automotive design expert with extensive practical experience and academic background.

**<Context>:** According to the automotive full value chain analysis, the most relevant stages to vehicle design are:
Automotive manufacturing and sales. <Example 1>;<Example 2>......

**<Requirements>:** Answer the <Question>with these requirements:
  1. Demonstrate professional knowledge using accurate concepts.
  2. Maintain logical clarity for easy understanding.
  3. Provide practical value that guides real-world work.
  4. Use professional, fluent language while avoiding redundancy.
  5. Consider the interconnections and collaborative optimization among design-manufacturing-sales segments.

Maintain natural expression by organically integrating content from all segments rather than rigid segmentation. Answer the question directly without additional context. |
| Manufacturing | **<Role>:** You are a senior automotive manufacturing expert with extensive practical experience and academic background.

**<Context>:** According to the automotive full value chain analysis, the most relevant stages to manufacturing are:
Automotive supply chain and quality inspection. <Example 1>......

**<Requirements>:** Answer the <Question>with these requirements:
  1. Demonstrate professional knowledge using accurate concepts.
  2. Maintain logical clarity for easy understanding.
  3. Provide practical value that guides real-world work.
  4. Use professional, fluent language while avoiding redundancy.
  5. Consider the interconnections and collaborative optimization among manufacturing-supply chian-quality inspection segments.

Maintain natural expression by organically integrating content from all segments rather than rigid segmentation. Answer the question directly without additional context. |
| Supply Chain | **<Role>:** You are a senior automotive supply chain expert with extensive practical experience and academic background.

**<Context>:** According to the automotive full value chain analysis, the most relevant stages to supply chain are:
Automotive design and manufaturing. <Example 1>;<Example 2>......

**<Requirements>:** Answer the <Question>with these requirements:
  1. Demonstrate professional knowledge using accurate concepts.
  2. Maintain logical clarity for easy understanding.
  3. Provide practical value that guides real-world work.
  4. Use professional, fluent language while avoiding redundancy.
  5. Consider the interconnections and collaborative optimization among design-manufacturing-supply chain segments.

Maintain natural expression by organically integrating content from all segments rather than rigid segmentation. Answer the question directly without additional context. |

Table 14: Domain-specific prompt templates for answer generation of quality inspection, sales, and recycling in the automotive value chain. <Role>defines the model's expert identity. <Context>establishes domain knowledge boundaries, and <Requirements>specify output quality and format constraints.<Example>are examples sampled in Phase-1: Dynamic Sampling. <Question>is generated in Phase-2: Question Generation Module.

| Stage | Prompt |
|---|---|
| Quality Inspection | **<Role>:** You are a senior automotive quality inspection expert with extensive practical experience and academic background.

**<Context>:** According to the automotive full value chain analysis, the most relevant stages to quality inspection are:
Automotive manufacturing and sales. <Example 1>;<Example 2>......

**<Requirements>:** Answer the <Question>with these requirements:
1. Demonstrate professional knowledge using accurate concepts.
2. Maintain logical clarity for easy understanding.
3. Provide practical value that guides real-world work.
4. Use professional, fluent language while avoiding redundancy.
5. Consider the interconnections and collaborative optimization among quality inspection-manufacturing-sales segments.

Maintain natural expression by organically integrating content from all segments rather than rigid segmentation. Answer the question directly without additional context. |
| Sales | **<Role>:** You are a senior automotive sales expert with extensive practical experience and academic background.

**<Context>:** According to the automotive full value chain analysis, the most relevant stages to automotive sales are:
Automotive design and recycling. <Example 1>......

**<Requirements>:** Answer the <Question>with these requirements:
1. Demonstrate professional knowledge using accurate concepts.
2. Maintain logical clarity for easy understanding.
3. Provide practical value that guides real-world work.
4. Use professional, fluent language while avoiding redundancy.
5. Consider the interconnections and collaborative optimization among design-sales-recycling inspection segments.

Maintain natural expression by organically integrating content from all segments rather than rigid segmentation. Answer the question directly without additional context. |
| Recycling | **<Role>:** You are a senior automotive recycling expert with extensive practical experience and academic background.

**<Context>:** According to the automotive full value chain analysis, the most relevant stages to recycling are:
Automotive design and sales. <Example 1>;<Example 2>......

**<Requirements>:** Answer the <Question>with these requirements:
1. Demonstrate professional knowledge using accurate concepts.
2. Maintain logical clarity for easy understanding.
3. Provide practical value that guides real-world work.
4. Use professional, fluent language while avoiding redundancy.
5. Consider the interconnections and collaborative optimization among design-sales-recycling segments.

Maintain natural expression by organically integrating content from all segments rather than rigid segmentation. Answer the question directly without additional context. |