# OpenReview forum: "MSCoRe: A Benchmark for Multi-Stage Collaborative Reasoning in LLM Agents"
_ICLR.cc/2026/Conference — ICLR 2026 Conference Withdrawn Submission_

### Official Review · Reviewer_fxk7 · 2025-10-22

**Soundness:** 2
**Presentation:** 3
**Contribution:** 2
**Rating:** 2
**Confidence:** 4

**Summary:**

The authors introduce MSCoRe, a novel benchmark comprising 126696 domain-specific QA instances spanning scenarios in automotive, pharmaceutical, e-commerce, and automotive energy sectors labelled with different difficulty.

**Strengths:**

- An important problem, multi-stage reasoning for automotive, pharmaceutical, e-commerce, and automotive energy sectors.

- The paper is easy to follow.

- The authors inject noise into data, including formatting errors, missing information, and semantic incompleteness, to assess robustness and interference resistance of models under adverse input conditions

- Use of dynamic sampling to generate questions from all the seed by linearly decreasing probability of selected seeds.

**Weaknesses:**

- All the dataset is synthetically generated from a small high quality seed to generate ~126K QA pairs, but no one can guarantee the correctness of the answers and the relevancy of the questions. You need a human in the loop to assess the questions and answers quality. This is the main reason for my score.

- typos (e.g., foucus → focus, Fotmulation->formulation)

- I am not sure why you need linearly decreasing probability for sampling. Using uniform distribution for sampling you should expect to have equally selected all the seeds.

**Questions:**

- Do all the questions that you generate require multi-stage reasoning?
- How do I ensure that the benchmark is reliable?

---

### Official Review · Reviewer_me67 · 2025-10-22

**Soundness:** 2
**Presentation:** 3
**Contribution:** 3
**Rating:** 6
**Confidence:** 2

**Summary:**

This paper introduces MSCoRe, a large-scale benchmark designed to evaluate multi-stage collaborative reasoning capabilities of LLM agents across four industrial domains: automotive, pharmaceutical, e-commerce, and automotive energy. The benchmark contains 126696 QA instances categorized into three difficulty levels (simple, medium, complex) based on the number of value-chain stages involved. The authors propose a structured three-phase pipeline—dynamic sampling, iterative QA generation, and multi-level quality assessment—to construct high-quality, domain-specific data. Comprehensive evaluations on 15 LLMs reveal that while commercial models perform best, all struggle significantly with complex, full-chain reasoning tasks and exhibit sensitivity to input noise.

**Strengths:**

1. Novel and Practical Focus: MSCoRe addresses a critical gap in existing benchmarks by emphasizing multi-stage collaborative reasoning—a realistic requirement in industrial workflows that prior benchmarks (e.g., MMLU, HotpotQA) largely overlook.

2. High-Quality Data Construction Pipeline: The proposed three-phase framework (dynamic sampling, refined prompt-based QA generation, and multi-level quality control with feedback loops) is well-designed and ensures both diversity and domain fidelity. The inclusion of expert-level professional assessment enhances data reliability.

**Weaknesses:**

1. Lack of Human Baseline: While a human-AI discrimination test is included, there is no quantitative comparison against human performance on the same tasks (e.g., human ROUGE scores or accuracy), making it hard to gauge how far models are from human-level reasoning.

2. Metric Limitations: Heavy reliance on ROUGE (and BLEU) may not fully capture reasoning quality, especially for complex, multi-stage answers where factual correctness, logical coherence, and cross-stage consistency matter more than lexical overlap.

3. Limited Analysis of Failure Modes: While error types are categorized in data generation, deeper qualitative analysis of why models fail on complex tasks (e.g., hallucination, stage omission, logical inconsistency) would strengthen the paper’s diagnostic value.

**Questions:**

1. Why were the problems from these four domains selected?

2. Why are only very small-scale open-source models used? Have experiments been conducted on larger-scale open-source models?

---

### Official Review · Reviewer_zKCF · 2025-10-27

**Soundness:** 2
**Presentation:** 3
**Contribution:** 2
**Rating:** 4
**Confidence:** 4

**Summary:**

This paper addresses the under-explored area of multi-stage collaborative reasoning in LLM-based agents. To bridge this gap, the authors introduce MSCoRe, a novel benchmark with over 126,000 domain-specific question-answering instances. A structured pipeline was developed to generate high-quality data for this benchmark. Evaluation of state-of-the-art LLMs revealed that while commercial models performed best, a significant performance gap remains between simple and complex tasks. The models' robustness was also found to be negatively affected by noisy data. The study concludes that MSCoRe serves as a valuable new resource for evaluating and improving multi-stage reasoning in LLM agents.

**Strengths:**

1. Overall, as a benchmark for evaluating the collaborative reasoning capabilities of LLMs in specialized domains, the authors have conducted extensive foundational work, involving data collection, data generation, quality assessment, and feedback optimization. This holds significant implications for evaluating LLM capabilities in collaborative reasoning scenarios.

2. Based on the constructed benchmark, the authors have performed extensive analyses, covering reliability analysis, few-shot learning analysis, noise stability analysis, and error analysis. The scope of the analysis is quite comprehensive.

**Weaknesses:**

1. The paper mentions real-world problems in the second paragraph of the Introduction, such as workflow tasks in industrial application scenarios, which often involve the connection of multiple steps, thus requiring collaborative reasoning to solve. However, the connection between multi-stage tasks and collaborative reasoning is not clearly explained. The emphasis of collaborative reasoning lies in the collaboration among multiple agents, which helps leverage economies of scale to solve complex problems. Whether multi-stage workflows are equivalent to large-scale collaboration among multiple agents requires more thorough explanation. For example, is a single agent sufficient to effectively complete an overall task composed of a series of simple tasks? Perhaps collaborative reasoning is just one of the ways to solve multi-stage tasks.

2. The paper mentions introducing three types of noise into the data to evaluate the model's reliability but does not explain why these three types were chosen. What is the connection between these noise types and multi-stage tasks or collaborative reasoning? The motivation for introducing noise remains unclear to readers.

3. The paper states that the problems in the constructed benchmark require not only domain expertise, but also collaborative reasoning to answer. However, the example given in Figure 1 is not representative enough to demonstrate that the dataset contains many problems that cannot be adequately answered by relying solely on expertise and must be addressed through collaborative reasoning. Such problems would better highlight the specificity of the constructed dataset. Can you provide examples where correct answers cannot be given using domain expertise alone but require collaborative reasoning?

4. Section 3.2, which introduces the data construction method, lacks a basic overview of the overall methodological framework. Why was this particular process adopted for data construction, and what is the basis for using such a process?

5. At the beginning of Section 3.2, a dynamic sampling method is introduced to obtain high-quality data for model learning, but the source of the data is not explained. Where does such a large amount of high-quality data, covering various professional fields and complete steps, come from?

6. Section 3.2 describes the question-and-answer generation process. However, since this benchmark primarily targets multi-step and collaborative reasoning tasks, it requires more comprehensive domain knowledge and broader step coverage. How is this data characteristic ensured during question generation? There seems to be a lack of explanation in this regard. Additionally, where does the contextual knowledge required for answer generation come from?

7. The Quality Assessment section for the generated data lacks detailed implementation descriptions. For example, how are Format Checks and Semantic Checks specifically carried out? Is there manual inspection, or is it entirely automated? If the process relies entirely on LLM generation and LLM evaluation, are there potential quality issues in the constructed data?

8. Regarding the performance differences of various LLMs on problems of different difficulties, as shown in Table 2, I believe radar charts could be used for visualization, as they would more intuitively display the strengths and weaknesses of different models. Additionally, the compared LLMs lack models specialized in specific professional domains. For instance, would LLMs adapted for the medical field demonstrate stronger collaborative reasoning capabilities when solving problems in the dataset compared to general-purpose models?

9. The paper's title emphasizes the evaluation of collaborative reasoning capabilities of LLM Agents, but from the content, the connection to agents does not seem prominent. There is no reflection of tool usage, memory, or other aspects of agents. Why is the emphasis placed on "Agent" rather than the LLM itself?

**Questions:**

See questions in weaknesses.

---

### Official Review · Reviewer_nGY8 · 2025-10-29

**Soundness:** 2
**Presentation:** 3
**Contribution:** 2
**Rating:** 2
**Confidence:** 3

**Summary:**

This paper proposes MSCoRe, a benchmark for evaluating LLMs’ multi-stage collaborative reasoning—addressing the gap of existing benchmarks that focus on isolated single-stage tasks. It includes 126,696 domain-specific QA instances across four sectors (Automotive, Pharmaceutical, E-Commerce, Automotive Energy Chains), uses a three-phase data pipeline (dynamic sampling, iterative Q&A generation, multi-level quality assessment), and categorizes tasks into simple/medium/complex levels. Experiments on 15 LLMs (e.g., GPT-4o, DeepSeek-R1) show commercial models perform best, but all degrade in complex tasks; robustness tests under three noise types (formatting, missing info, semantic incompleteness) reveal negative impacts on performance.

**Strengths:**

1. Focuses on multi-stage collaborative reasoning (critical for real industrial workflows like automotive design-manufacturing-recycling)
2. Replaces uniform random sampling with dynamic sampling to mitigate data imbalance and enhance dataset diversity.

**Weaknesses:**

1. The manuscript repeatedly refers to "LLM agents" but only evaluates multi-stage reasoning through QA tasks—failing to test core agent capabilities (e.g., tool use, environment interaction, dynamic decision-making based on stage feedback). This renders the term "LLM agent" inconsistent with the benchmark's actual scope.
2. While appendices detail prompt templates, there are no actual QA examples for simple/medium/complex tasks. Readers cannot verify how task difficulty translates to real outputs or assess if the benchmark truly captures cross-stage dependencies.
3. Over-reliance on ROUGE and BLEU—metrics based on lexical overlap—fails to capture semantic similarity; for instance, answers with equivalent multi-stage reasoning but different wording may receive low scores. The omission of semantic metrics like BertScore undermines the accuracy of reasoning quality evaluation.
4. As a general-purpose benchmark, the paper covers a limited range of domains, which may restrict its applicability and generalizability across diverse tasks or fields.

**Questions:**

see weaknesses

---

### Note · Authors · 2025-11-17

I have read and agree with the venue's withdrawal policy on behalf of myself and my co-authors.